

# Imminent reversal of the residual flow through the Marsdiep tidal inlet into the Dutch Wadden Sea based on multiyear ferry-borne ADCP observations

Johan van der Molen[1], Sjoerd Groeskamp[1], Leo R.M. Maas[2]

[1] Royal Netherlands Institute for Sea Research (NIOZ), P.O. Box 59, 1790 AB Den Burg (Texel), The Netherlands.
[2] Institute for Marine and Atmospheric Research (IMAU), Utrecht University, Princetonplein 5, 3584 CC Utrecht, The Netherlands

*Correspondence to*: Johan van der Molen (johan.van.der.molen@nioz.nl)

**Abstract.** The Dutch Wadden Sea is a UN World Heritage Site connected to the North Sea by multiple tidal inlets. Although there are strong tidal currents flowing through these inlets, the magnitude and direction of the residual circulation in the western Dutch Wadden Sea is important for sediment, salinity and nutrient balances. We found that the direction of this residual flow is reversing.

This residual circulation has been the subject of various studies since the 1970's, in which substantially different net volume fluxes were presented. As driving mechanisms differences in tidal conditions in the main inlets, tidal rectification, and meteorology were identified. Here we analysed almost 13 years of ADCP observations collected on the ferry crossing the Marsdiep tidal inlet in the Dutch Wadden Sea since 2009. The results are combined with earlier investigations covering the period 1998-2009. We find a significant trend in the magnitude of the residual volume flux, with decreasing export to the North Sea, and with occasional imports observed in recent years. We infer that this trend is related to changes in tides in the North Sea, which are caused by increased strength and duration of stratification in response to global warming. With warming projected to continue, we expect the residual flow in the Marsdiep to continue to reverse to full inflow within the current decade, with potential knock-on effects for the sediment balance and ecosystem of the western Wadden Sea.

## 1. Introduction

The Marsdiep (Figure 1) is the largest and western-most tidal inlet in the Dutch Wadden Sea, which is a UN World Heritage Site, an important nursery area for North Sea fish, and a vital staging area and food source for migratory birds. The inlet separates the island of Texel from the mainland and is connected to the Vlie inlet further north by back-barrier channels, allowing for a net residual water circulation through the western Dutch Wadden Sea. This residual circulation carries nutrients, eggs and larvae, and influences the regional salt and sediment balances. In 1998, flow velocity measurements were started using Acoustic Doppler Current Profilers (ADCPs) mounted on the TESO (Texels Eigen Stoomboot Onderneming) ferry across the Marsdiep, which are still on-going. One of the objectives was to quantify the net residual flow through the inlet. However, various studies of the data, each based on periods of time up to several years, provided differing results for the magnitude of the residual flow. Since these earlier studies, more than a decade of additional data has been collected, and the current length of this unique data set allows for a more rigorous analysis, including determination of decadal trends. This



paper presents an analysis of decadal trends and patterns in residual flows in the Marsdiep inlet based on these long-term ADCP data. The remainder of this introduction provides an overview of the hydrography and morphology of the study area, including a summary of the earlier estimates of residual flow.

The largest depth of the Marsdiep is over 40 m, and the width is approximately 4 km. Tides in the Marsdiep are semi-diurnal, with a range of about 1.37 m (Nieuwhof & Vos, 2018). The tides are progressive in character, and tidal current speeds can reach 1.8 ms$^{-1}$ and are strongly asymmetrical (Buijsman & Ridderinkhof, 2007). Fresh water is discharged into the western Wadden Sea through sluices, with a mean flux estimated at 450 m$^3$s$^{-1}$ (Ridderinkhof, 1990), this can result in weak surface to bottom salinity gradients (Zimmerman, 1976a) and stratified conditions that carry internal waves (Groeskamp et al., 2011).

Residual currents are directed into the inlet in the southern part of the inlet, and out of the inlet in the northern part, as a result of a suspected tidally driven residual eddy (Zimmerman, 1976b; Ridderinkhof, 1988). Tides are flood-dominant in the southern two-thirds and ebb-dominant in the northern one-third of the inlet (Buijsman & Ridderinkhof, 2007). The tidal basin of the Marsdiep inlet has an average tidal prism of 0.99*10$^9$ m$^3$ with a standard deviation of 0.18*10$^9$ m$^3$ (Duran-Matute et al., 2014).

Volume transports at peak tidal currents are 5-10*10$^4$ m$^3$s$^{-1}$ (Buijsman & Ridderinkhof, 2007; Nauw et al., 2014). Over-all residual flows are from the Vlie inlet to the Marsdiep, and assumed to be driven by differences in tidal amplitudes and phases between the two inlets, but display substantial wind-driven variability (Ridderinkhof, 1988; Buijsman & Ridderinkhof, 2007; Duran-Matute et al., 2014). As a result, reported residual flows through the Marsdiep vary, depending on the period considered (Table 1). Despite the generally outward direction of the residual flow, time lags and non-linearities ensure a residual import of suspended particulate matter (SPM) of 7-11 Mton yr$^{-1}$ (Nauw et al., 2014).

The sea-bed morphology of the Marsdiep inlet is dominated by sand waves. In the northern half of the inlet, the waves are asymmetrical-trochoidal with heights and lengths of about 2 and 165 m, respectively and are subject to seasonal variations which may be related to temperature-driven changes in water viscosity, while in the southern half sand waves are progressive with heights and lengths of about 3 and 190 m, respectively (Buijsman & Ridderinkhof, 2008a,b). The sand waves migrate in the flood direction with speeds up to 90 m yr$^{-1}$. In the southern part, sand-wave migration and bedload transport are in the flood direction, and slightly more than half of the bedload transport is caused by tidal asymmetry, while the rest is caused by residual currents. In the northern part, sediment transport rates are opposite to the sand-wave migration direction suggesting that suspended load processes are dominant, in agreement with their seasonal variability (Buijsman & Ridderinkhof, 2008b).

The tidal basin of the Marsdiep inlet was truncated by the damming of the 'Zuyderzee' by the closure dam 'Afsluitdijk' in 1932 (creating the current Lake IJssel), which caused a strong tidal and morphodynamic response. Although the morphology of the western Wadden Sea is still adjusting, the resulting import of sand has reduced to small values since 1980 (Elias et al., 2012; Wang et al., 2012), but with ongoing import of mud (Colina Alonso et al., 2021). The present small rates of sand import suggest that, in terms of volume, the system has reached a new dynamic equilibrium and is now mostly responding to



other forcings such as climate-change, sea-level rise and beach nourishments (Wang et al., 2012). Alongside these changes in the tidal basin, the ebb-tidal shoal Noorderhaaks has grown, and the seaward end of the inlet channel has shifted closer to the mainland coast (Elias et al., 2012), a process that is likely still on-going.

## 2. Methods

### 2.1 ADCP observations

Two Teledyne RD Instruments Workhorse Monitor 1200 kHz ADCPs (www.teledynemarine.com/workhorse-monitor-adcp) were mounted on the roll-on, roll-off ferry. One at each end of the ship, at 4.5 m below the sea surface. The ferry has identical bow and stern design and reverses sailing direction at each crossing (Figure 1). In this configuration, one instrument is at the front of the ferry at any one time and undisturbed by the wake of the ship. Each instrument has four beams at angles of 20º. Data were recorded in 50 bins of 0.5 m width. Further instrument settings are given in Table 2. In addition, both ends of the ferry were fitted with a differential GPS (JRC JLR-21/31), and a gyroscopic compass (Alphatron Alphaminicourse) to record position and orientation.

The ADCP, GPS and directional data were combined and stored by an on-board computer during the day, and transmitted to a shore-based server when the ferry was in port at night. Recorded variables included navigational variables, water temperature, depth below the instrument for each beam, two orthogonal horizontal and one vertical velocity components for each bin, signal attenuation along each beam for each bin. The horizontal velocities were converted to geo-referenced eastward and northward velocity components using the navigational data following the correction method for heading and tilt described by Joyce (1989), see also Buijsman and Ridderinkhof (2007). Water-depth and bin-depth measurements were automatically corrected by the instrument for variations in sound speed propagation using the instrument-mounted temperature sensor and a user-specified salinity of 28. Water depths were averaged over the four beams and corrected to depth below surface using the draught of the ship. The data also include the spike identification velocity (or 'error velocity') for each bin, which is a measure of the variation in the result when velocity is computed using different combinations of three of the four beams, and is hence a measure for the reliability of the observations.

Typically, sailings started at 6:00, with an hourly outward-bound schedule from Texel and return journeys starting on the half hour from Den Helder (Figure 1). Crossings typically took about 20 minutes. The last outward-bound sailing of the day started at 21:00, resulting in 32 single crossings per day. The sailing schedule was adjusted by the ferry operator in summer to account for daylight savings time. This schedule was maintained on most days of the year, with the exception of Christmas Day, New Year's Day, scheduled maintenance periods in January and November, and occasional unscheduled maintenance or extreme weather events. Occasionally, instrument failure also led to reduced data return. From March 2009 to July 2016 the ADCPs were mounted on the 'Dokter Wagemaker', and from July 2016 to present on the 'Texelstroom'. The setup on the 'Dokter Wagemaker' was configured similarly as described above, but with a single GPS midships (Furuno SC110, www.furunousa.com/en/products/sc110), and a single gyroscopic compass included in the GPS.





### 2.2 Data processing

**2.2.1 Quality Assessment**

Quality flags were added to the depth-resolved variables using a threshold value for the spike identification velocity. For the values of the Quality Assessment (QA) flags, we used designations defined by SeaDataNet (49 = good value, 50 = probably good value, 52 = bad value, www.seadatanet.org/Standards/Data-Quality-Control). To define the threshold values, we inspected transects recorded by the ADCP mounted on the rear of the ship. Such records contain clearly visible anomalous

velocities when the ADCP signal is disturbed by the wake of the thrusters of the ship. Such wake-affected velocities were adequately identified using a spike identification velocity threshold of 0.15 ms$^{-1}$, and nearly always for a spike identification velocity threshold of 0.2 ms$^{-1}$. Hence, data points corresponding with a spike identification velocity threshold less than 0.15 ms$^{-1}$ were flagged as 'good value', between 0.15 ms$^{-1}$ and 0.2 ms$^{-1}$ as 'probably good value', and greater than 0.2 ms$^{-1}$ as 'bad value'. As such disturbances in the observations tended to extend throughout the water column, all bins in the vertical at

locations with more than 30% 'bad values' were flagged as 'bad value'. The same thresholds were assumed to hold for the ADCP mounted on the front of the ship, and seem appropriate as it allows for errors of up to about 15% of typical maximum currents. This choice is however somewhat subjective, and other threshold values could be set for different applications, e.g. when individual transects are studied. In the remainder of this paper, both 'good values' and 'probably good values' were included.

**2.2.2 Latitudinal gridding of each crossing**

Because of variations in the speed of the ferry, different crossings resulted in varying amounts of data with location-dependent data density. Moreover, as the data from both front and rear ADCPs typically contain QA-related gaps, not a single crossing resulted in continuous coverage of the cross section. To obtain regular data, use the results of both ADCPs, and obtain maximum coverage, each transect was subdivided into 100 equidistant latitudinal intervals of 42.74 m, while

retaining the longitudinal positions and vertical gridding. The data from both ADCPs were collected in each resulting latitude-depth grid cell, and averaged. For each latitude-depth grid cell, a standard deviation was calculated to provide an uncertainty estimate for the mean value of each grid cell. Water depths were similarly averaged per latitudinal grid interval. These data were used to calculate cross-section aggregated averages and trends (Section 2.2.3). To calculate cross-section resolved patterns (Section 2.2.4), similar gridding was carried out, but after first projecting each observed column of the

curved transect onto a straight central north-south transect subdivided into the same latitudinal intervals. This projection was carried out by moving each column in the opposite direction of the current measured at the surface until it met the straight central transect. Resulting data on the central transect were averaged for each latitudinal interval.

**2.2.3 Time series and trends of cross-sectionally aggregated depths and flows**

Cross-sectionally averaged depths and eastward cross-sectionally integrated volume fluxes were calculated for each

latitudinally gridded crossing to construct time series to analyse for decadal trends.

Cross-sectionally averaged depths contain some variation related to longitudinal position and associated sea-bed morphology (e.g., Buijsman & Ridderinkhof, 2008a,b), but these will only contribute to the uncertainty of the trend estimates (see



below), and not introduce a bias. Cross-sectionally averaged depths were calculated by averaging over the latitudinal grid intervals. Annual mean depth profiles along the cross section were also plotted to illustrate the temporal changes.

As the transects cover the entire cross section and both ends of the transects have nearly identical longitudinal coordinates (resulting in only a net north-south displacement of the ship for each crossing), eastward cross-sectionally integrated volume fluxes represent the full volume flux through the Marsdiep. Eastward cross-sectionally integrated volume fluxes were calculated by first copying the eastward flow velocities of the top grid cell to the water surface, then multiplying by the surface area of each cross-sectional grid cell, and finally adding all cross-sectional grid cells.

Tidal harmonic analysis (Doodson, 1921) was applied to the time series as a whole, estimating the amplitudes of the 21 main tidal constituents, including nodal corrections, their Greenwich phase, and the mean residual component. The time series was also split up into yearly intervals, with tidal harmonic analysis applied to each individual year. Buijsman & Ridderinkhof (2007) established that the largest 15 constituents represent 96% of the tidal variance. Yearly intervals were used to enable detection of trends in harmonic constituents, and to facilitate comparison of the yearly mean residual volume fluxes to earlier

estimates. To obtain trends in the residual cross-sectional depth changes and volume fluxes, the time series as a whole were de-tided by subtracting a tidal time series reconstructed from the harmonic analyses. Decadal trends were calculated for the de-tided time series as the slope of a straight line estimated using a least-squares fit. To estimate trends in the tidal harmonic constituents, regression lines were fitted through the yearly amplitudes and phases. Here, we only present and discuss variables with an estimated slope that exceeds two times the standard error of the fit (corresponding to a 95% confidence

interval).

To test the hypothesis (Ridderinkhof, 1988; Buijsman & Ridderinkhof, 2007; Duran-Matute et al., 2014) that the variations in residual flow are mainly wind-driven, a scatter plot was constructed of the annual residual transports as a function of the annual mean eastward wind-speed component derived from the ECMWF ERA5 reanalysis (C3S, 2019).

### 2.2.4 Cross-section resolved temporal mean patterns and trends

To calculate cross-section resolved temporal mean patterns and trends, the latitudinally gridded data were first corrected for water-level changes such as tides using the bed level of the first couple of latitudinal grid intervals of the transect on the Texel side as a reference. Then, annual and decadal means and trends were calculated for the residual velocity components for each latitude-depth grid cell in the same way as for the transports: by performing a harmonic analysis, de-tiding the time series, and calculating a linear least-squares fit through the residuals. To present the results, contour plots of the residual

velocities and trends, as well as line graphs of the depth- and cross-section averaged residual velocities were constructed.

## 3.  Results

### 3.1  Example of Quality Assessment

As an example of the QA procedure, eastward velocities measured on the transect sailed on 7 April 2021, 6:00 local time are given (Figure 2). The left two panels show the uncorrected eastward velocities for the front (a) and rear (c) ADCP, and the

panels on the right (b,d) the corresponding QA-ed velocities. In Figure 2c, the data affected by the ship's propulsion system





are clearly visible, and were effectively removed in **Error! Reference source not found.**d. The QA procedure also removed some data from the front ADCP (Figure 2b). To further illustrate the procedure, the spike identification velocity for the second ADCP bin of both instruments was plotted along the transect, including the thresholds (Figure 3).

### 3.2 Example of latitudinal gridding

An illustration of the latitudinal gridding of the QA-ed data of both ADCP's, combined for the same date and time shows a coherent profile without data gaps (Figure 4a). Despite strong variation in the number of data points per grid cell (Figure 4b) due to a combination of changes in vessel speed and gaps in the QA-ed data (Figure 2), standard deviations showed no obvious pattern over the cross-section (Figure 4c) (except near the bottom as the two ADCP's, which are a ship's distance apart, may follow slightly different trajectories and encounter different sea-bed topography), suggesting uniform quality of
the eastward current speeds constructed in this way for the cross-section.

### 3.3 Time series of cross-section aggregated averages and trends

Time series of cross-section averaged residual depth (Figure 5a) showed an average of 18.4 m and a shoaling trend of 1.88 cm yr$^{-1}$ with a standard error of 0.04 cm yr$^{-1}$. Similarly, the cross-sectionally integrated volume flux (Figure 5b) showed an average outflow that changed from around 1000 m$^3$ s$^{-1}$ in 2009 to around zero in recent years, with a reducing trend of 98 m$^3$
s$^{-1}$ yr$^{-1}$, and a standard error of 13 m$^3$ s$^{-1}$ yr$^{-1}$, that continued compared with values reported for earlier years. Most notably, the volume flux is transitioning to positive values (net inflow) in recent years. Finally, the amplitude of the O1 tidal constituent of the volume flux (Figure 5c) more than doubled in the period considered, with a trend of 249 m$^3$ s$^{-1}$ yr$^{-1}$ and a standard error of 68 m$^3$ s$^{-1}$ yr$^{-1}$. This was the only tidal constituent in the ADCP data with a consistent trend exceeding twice the standard error.

Plotting the annual residual eastward volume fluxes as a function of the annual mean eastward windspeed component did not suggest an evident correlation (not shown).

The annual-averaged depths along the transect (Figure 6) revealed the infilling of a substantial depression in the southern flank of the inlet around 52.975 N, as well as an increase in the maximum depth along the transect between 52.990 N and 52.985 N, but no major changes in the cross-sectional geometry. The apparent shift in the flank of the channel near Texel
after 2017 coincides with the change-over of the instrumentation from the Dokter Waagemaker to the Texelstroom, and is likely an artifact, potentially related to a slight change in approach route related to differences in handling of the ships, which differ considerably in size. Note that the profile of 2009 is less reliable than the others as only data from the later part of the year were available.

### 3.4 Cross-section resolved temporal mean patterns and trends

The 13-year residual eastward velocities showed outflow in the northern part of the inlet, with highest values close to the coast of Texel in the upper part of the water column, and inflow in the southern part of the inlet, with higher values in the lower part of the water column (Figure 7a). Both inflow and outflow appear to consist of two kernels. The trends in the



residual eastward velocity show a vertically banded structure of alternating increases and decreases, with the strongest band in the southern part of the inlet (Figure 7c). A striking feature was a strongly negative trend near the bottom in the southern

part of the inlet. Standard errors of the trend estimates were small, except near the surface and the bottom, where it increased to 0.002 ms$^{-1}$yr$^{-1}$ due to the fluctuating water levels and local variations in depth associated with individual transects and morphodynamic changes affect the estimates.

The northward velocities were positive near the surface in the southern part of the inlet, and near the bottom in the northern part of the inlet (Figure 7b). They were negative elsewhere, with weakest negative values in the central part of the inlet. This

pattern suggests a double gyre in the plane of the transect, but this could not be confirmed as the vertical velocities recorded by the ADCPs were affected by the flow induced by the forward motion of the ship. The trends in the northward velocity reflected the vertically banded structure of the trends in the eastward velocities, but also indicated a weakening of both the positive and negative residual velocities, suggesting a weakening of the suspected double gyre pattern (Figure 7d). The standard errors of the trend estimates were similar to those of the eastward velocity.

Depth- and cross-section averages of the residual eastward velocities provide another way of presenting the main patterns, and also allow for a depiction of the interannual variations. The depth-averages of the 13-year residuals again showed the outflow in the northern part, and the inflow in the southern part (Figure 8a, thick black line). The individual years clustered relatively close to this line, with the largest variations near the maximum residual inflow in the southern part of the inlet. The cross-section averaged values of the 13-year residual eastward velocity showed outflow in the top 10 m of the water column,

and inflow in the 15 m below that (Figure 8b, thick black line), suggesting a component of estuarine circulation to the flow. Interannual variations were largest for the outflow velocities.

## 4. Discussion and conclusions

### 4.1 Cross-section resolved residual flow patterns and trends

The overall cross-sectional pattern of inflow in the southern part of the inlet and outflow in the northern part of the inlet,

with a weak estuarine circulation component in the vertical corresponds with the results of earlier investigations (Zimmerman, 1976a,b; Ridderinkhof, 1988; Buijsman & Ridderinkhof, 2007). Huijts et al. (2009) used a semi-analytical model of a schematised curved tidal channel to investigate the cross-sectional patterns of residual flows in the stream-wise and cross-channel directions caused by tidal rectification, along-channel density gradients, along-channel winds, and river discharge. Their pattern for streamwise residual flow induced by tidal rectification corresponds well with the pattern of

eastward residual flow in the Marsdiep, including the double inflow/outflow kernels, which only occur for the tidal rectification component. Their pattern for cross-channel residual flow by tidal rectification shows a double gyre pattern like that of the northward residual currents in the Marsdiep, but with reversed direction. The causes of this difference are not immediately clear, but the geometry of the Marsdiep is much more complex than the schematized curved channel in the model of Huijts et al. (2009) (Figure 1): i) the main channel is not aligned with the coasts and the angle is different at either

side; ii) just seaward of the transect, the main channel curves in the other direction, and on the basin side the main channel is





more or less straight. Moreover, the ferry transect slants across the main channel at a fairly large angle, the exact value of which is difficult to estimate because of these changes in curvature. The direction of the gyre pattern corresponds with that observed by Nunes and Simpson (1985) during the flood phase of the tide in an estuary in Wales (UK), and explained by strong friction with the sides of the channel. Cui et al. (2018) observed this pattern as well in an estuary in the Gulf of

Mexico, where they also observed the reverse pattern during the ebb phase. So it is possible that, in a flood dominant estuary like the Marsdiep, the flood-phase pattern is expressed in the long-term mean as we see in the observations presented here. As Huijts et al. (2009) used a purely sinusoidal tide superimposed on a river runoff, effectively creating ebb dominance in their model, this reasoning may explain the discrepancy if this effect dominates over the influence of geometric differences. Hence, the correspondence of the observed eastward residual flow pattern with the modelled streamwise residual flow

pattern induced by tidal rectification is a strong indication that this mechanism is dominant in shaping the cross-sectional residual flow patterns in the Marsdiep.

The trends in the cross-sectional residual flow pattern suggest a slight intensification of both the outflow in the northern part of the inlet and of the inflow in the southern part of the inlet, in particular in later years, with the inflow increasing more than the outflow. There were slight latitudinal shifts in the double kernel pattern, but these do not seem to have a systematic

direction. There was also a local response to the infilling of a depression at 52.975N. These trends suggest changes in the tidal rectification process, possibly related to larger-scale morphodynamic changes such as the on-going infilling, sea-level rise, and channel migration in the Wadden Sea and in the ebb-tidal delta (Wang et al., 2012; Elias et al., 2012). Slight changes in tides may also be involved, as is suggested by the observed trends in O1 currents and flows. These could be related to bathymetric changes (Benninghoff & Winter, 2019; Jacob & Stanev, 2021; Colina Alonso et al., 2021) and/or sea-

level rise (Wachler et al, 2020) that may alter the resonance characteristics of the basin. Larger-scale changes in tides in the North Sea likely also play a role: these resulted in small increases in tidal range near the Marsdiep from 2010-2015, but also in differences in trends in tidal range between the Marsdiep and the Vlie inlets between 1958 and 2014 (Jänicke et al., 2021). They identify two components driving the changes in tides in the North Sea: a large-scale barotropic component that involves the north Atlantic Ocean, and a regional baroclinic component that relates to stronger stratification in the southern

North Sea. Such a baroclinic component may favour changes in diurnal constituents as a result of the day/night heating/cooling cycle. Tide gauge data most likely offer a more accurate way to analyse trends in tides in the study area than the ADCP data analysed here as they are more accurate, have higher sampling frequency, measure continuously, and cover a wider area. However, such a study is beyond the scope of this paper.

## 4.2 Trends in cross-section aggregated residual flows

We found a significant trend in the cross-section aggregated residual flow in the Marsdiep tidal inlet, suggesting an imminent reversal from net outflow to net inflow. Ridderinkhof (1988) showed that the outflow at that time was related to a residual circulation entering through the Vlie inlet, and driven by differences in amplitudes and phases of the tides between the two inlets. We demonstrated that wind forcing, suggested in earlier studies to influence residual flows (Buijsman &



Ridderinkhof, 2007; Duran-Matute et al., 2014), is not correlated to the trend observed in the ADCP data. The most likely
process driving the trend in residual flows is the baroclinically derived change in the difference in tides between the
Marsdiep and Vlie inlets caused by climate change identified by Jänicke et al. (2021). Indeed, plotting the annual mean
cross-section aggregated residual eastward flow as a function of local annual averaged air temperature derived from the
ECMWF ERA5 reanalysis shows a correlation (Figure 9). Although this of course does not indicate a causal relationship, it
is in line with the stratification mechanism proposed by Jänicke et al. (2021). As a result, with warming trends expected to
continue, a permanent reversal of the residual flow followed by a strengthening net inflow is likely to occur in the next
decade(s). Similar changes may happen in other multiple inlet systems boardering temperature-stratified seas elsewhere in
the world.

### 4.3  Potential consequences of a reversal in residual flow

Reversal of the residual flow in the Marsdiep will be linked with a reversal and potential changes in the residual flow
patterns in the western Wadden Sea. This may influence physical factors such as temperature and salinity distributions, and
may also increase the import of fine suspended sediment through the Marsdiep inlet, while reducing that through the Vlie
inlet. Moreover, it may change pathways of nutrient supply and transport of passive propagules (eggs and early stage larvae
of marine organisms). Such changes may cascade up through the Wadden Sea ecosystem, that is already influenced by
anthropogenic pressures and more direct effects of climate change. However, as the residual flows are much smaller than the
tidal flows, and even heterogeneous within the Marsdiep inlet itself, it is not possible to infer such effects from this study and
further work is needed.

### 4.4  Suggestions for further research

Further work is needed to fully understand the causes, significance and effects of reversal of the residual flow in this multi-
inlet tidal embayment. Next steps could include detailed analysis of the local tide gauge data to refine understanding of the
driving forces, model projections of climate-driven trends in North Sea tides, model studies detailing the processes driving
the residual circulation in the Wadden Sea and projecting future changes as well as implications for the regional sediment
balance, and model studies projecting changes in the ecosystem in response. Existing observations of ecosystem variables
and components should be examined, and wider observational campaigns could be initiated to gain further understanding,
underpin models and observe predicted effects.


**Data availability**

The   TESO   ADCP   data   will   be   made   available   through   the   NIOZ   Data   Archiving   System
(https://www.nioz.nl/en/research/dataportal/das).

**Author contributions**



JvdM designed the study, processed the data, carried out the analysis and interpretation, and wrote the manuscript. LM and SG contributed to several revisions. All authors supervised Erin Lejeune and Mariana de Botton.

**Competing interests**

The authors declare that they have no conflict of interest.

**Acknowledgements**

Many people contributed to collecting the TESO ADCP time series data, a collaborative effort of the Royal Netherlands Institute for Sea Research (NIOZ) and the Royal TESO N.V. ferry company for more than two decades, and we want to

extend our thanks to them, although not all can be named. Herman Ridderinkhof initiated the observations. Frans Eijgenraam and Eric Wagemaakers developed, installed and maintained the adaptations to the ferries, the instrumentation and the software. Numerous MSc students, PhD students and postdocs have worked on the data, and some of their papers are referenced here. In recent years, Erin Lejeune and Mariana de Botton, both funded through Utrecht University – NIOZ work experience placements, worked on modernising the automated data postprocessing that made this paper possible.

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



**Table 1. Previously reported residual volume fluxes through the Marsdiep inlet.**

| Years | Flux [$10^3$ m$^3$s$^{-1}$] | Type | Reference |
|---|---|---|---|
| 1998-2002 | -2.91 | ADCP, period averaged | Buijsman & Ridderinkhof (2007) |
| 2003-2005 | -1.1 | ADCP, period averaged | Nauw et al. (2014) |
| 2009 | -0.5 | model, annual average | Sassi et al. (2016) |
| 2009-2010 | -0.6 to -0.7 | model, 'typical conditions' | Duran-Matute et al. (2014) |

**Table 2. Instrument settings.**

| Parameter | Value | Units |
|---|---|---|
| prof_mode | 1 | - |
| coord_sys | earth | - |
| orientation | down | - |
| beam_pattern | convex | - |
| pings_per_ensemble | 1 | - |
| blank | 1 | - |
| avg_method | time | - |
| avg_interval | 1 | s |
| magnetic_var | 0 | ° |
| compass_offset | 0 | ° |
| xducer_misalign | 45 | ° |
| intens_scale | 0.43 | dB |
| absorption | 0.382 | dB m$^{-1}$ |
| salinity | 28 | - |
| use_pitchroll | yes | - |
| bin1_dist | 1.52 | m |
| xmit_pulse | 0.5 | m |






**Figure 1. Study area with a selection of ferry crossing trajectories. Contours are depths below mean sea level in m. Insets: wider area with arrows pointing to i) the study area in the Marsdiep inlet, and ii) the Vlie inlet; the blue line is the closure dam 'Afsluitdijk'.**





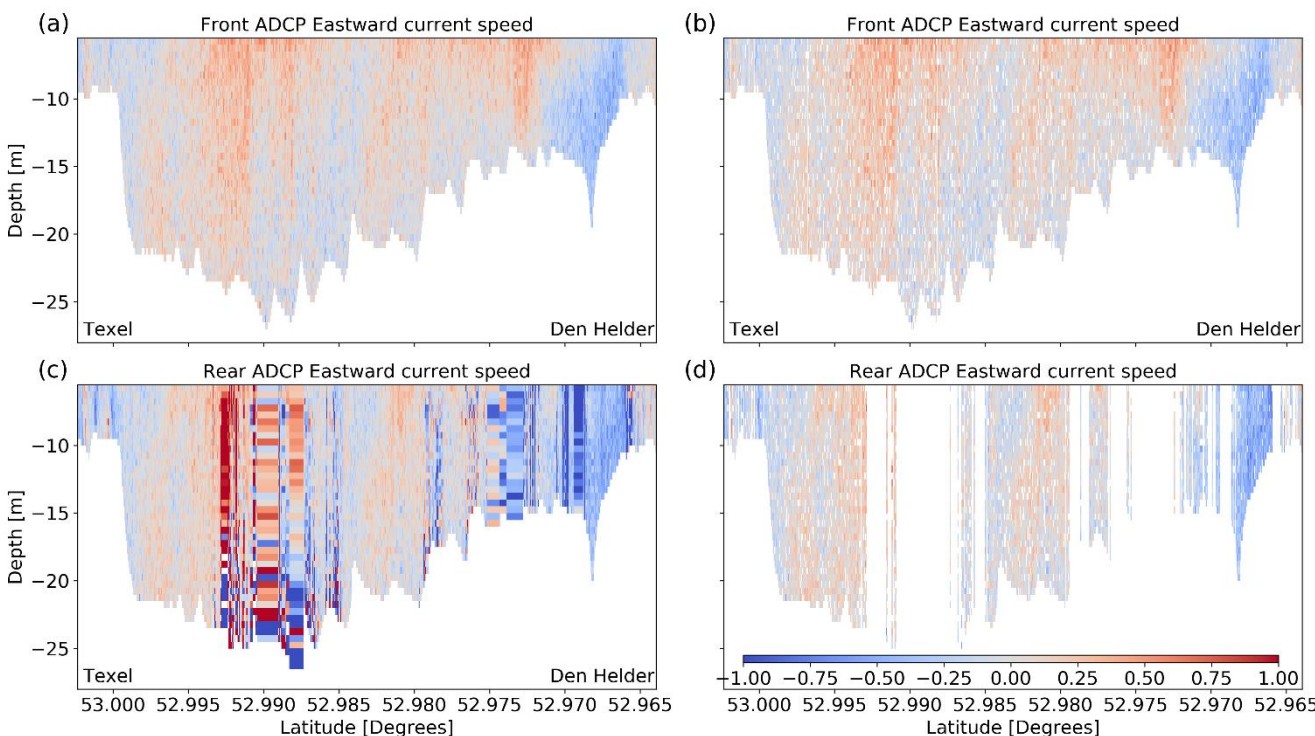

**Figure 2. Example of QA procedure, 7 April 2021, 6:00 departure, sailing from Texel to Den Helder. a) original and b) 'good' and 'probably good' data for the front ADCP. c) and d): idem. for the rear ADCP. Velocities in ms⁻¹.**

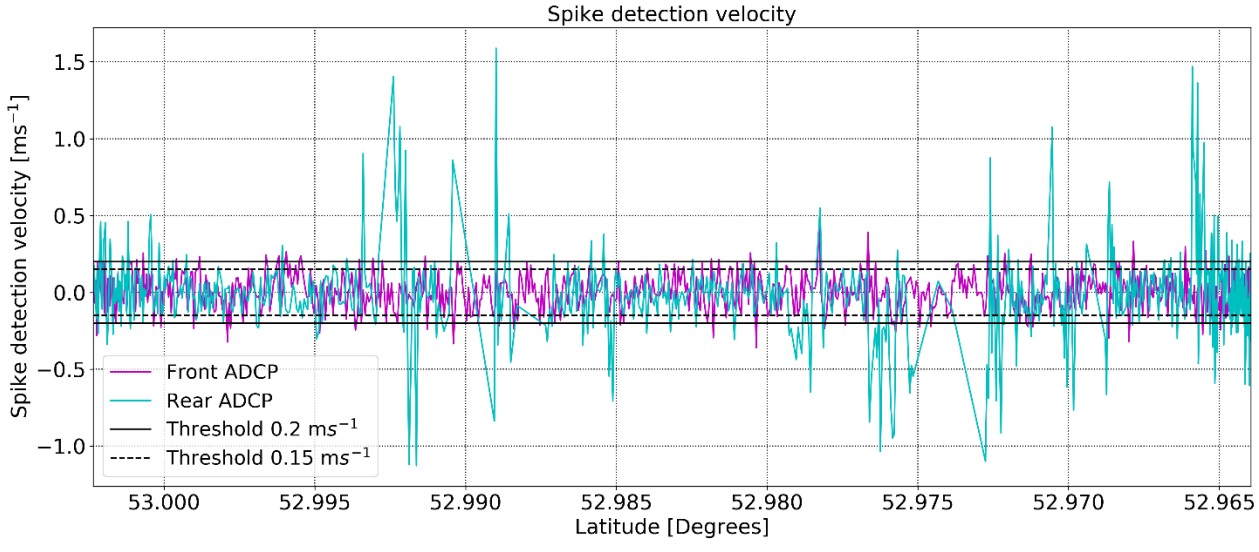


**Figure 3. Example of spike detection velocity compared to QA thresholds, second bin below the vessel, 7 April 2021, 6:00 departure.**





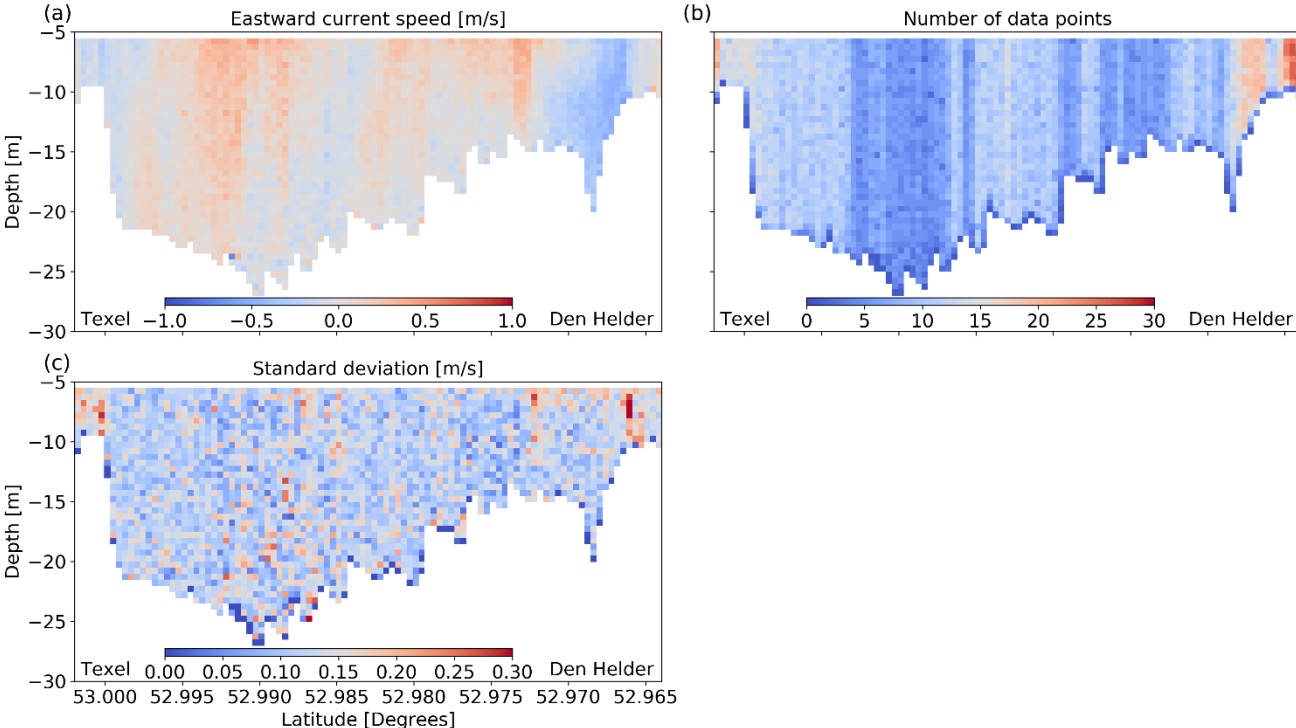

**Figure 4. Example of transect based on latitudinally gridded, QA-ed data from front and rear ADCP (Figure 2b, d), 7 April 2021, 6:00 departure. a) Resulting mean gridded eastward current speed, b) number of data points per grid cell, c) standard deviation per grid cell.**



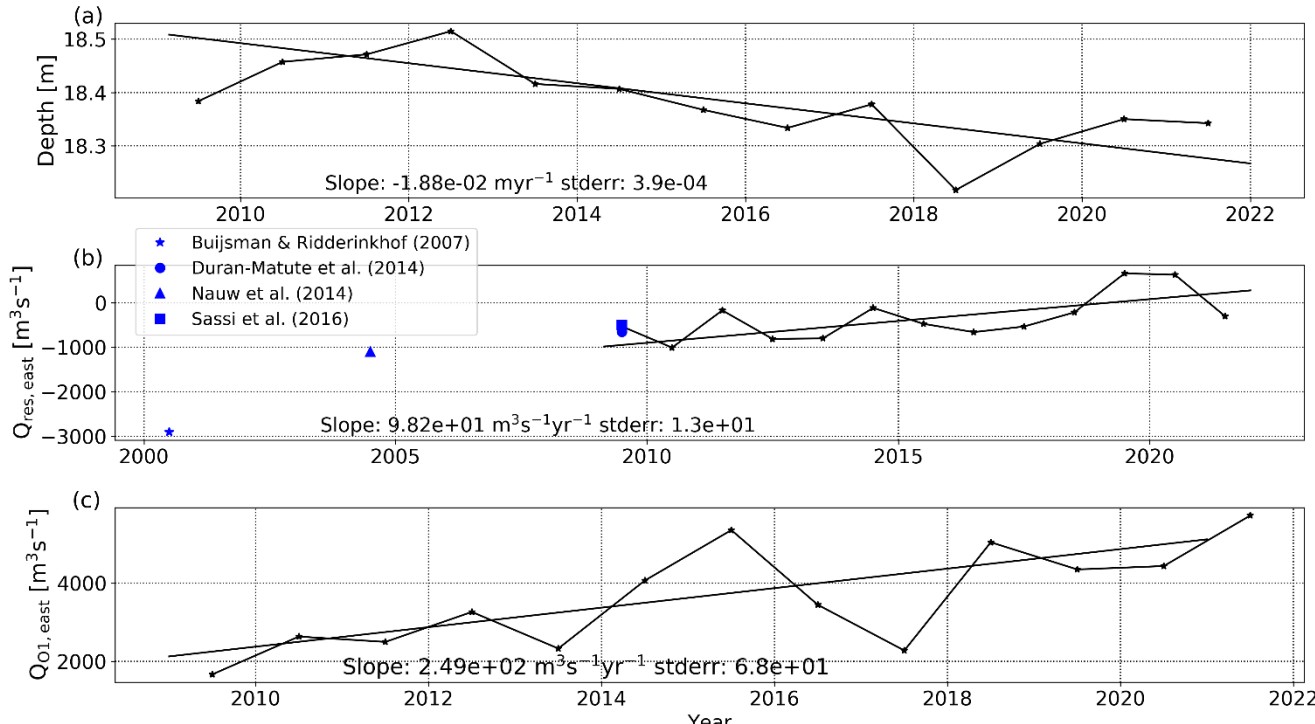

**Figure 5. Observed trends after applying harmonic analysis. A) annual residual cross-sectionally averaged depth, trend line based on the de-tided time series b) annual residual cross-section integrated eastward volume flux including values reported earlier, trend line based on the de-tided time series, c) annual O1 amplitude of cross-section integrated eastward volume flux, trend line based on the annual amplitudes.**




**Figure 6. a) Annual averaged depths along the transect, and b) the annual depth anomaly with respect to the 13-year mean.**




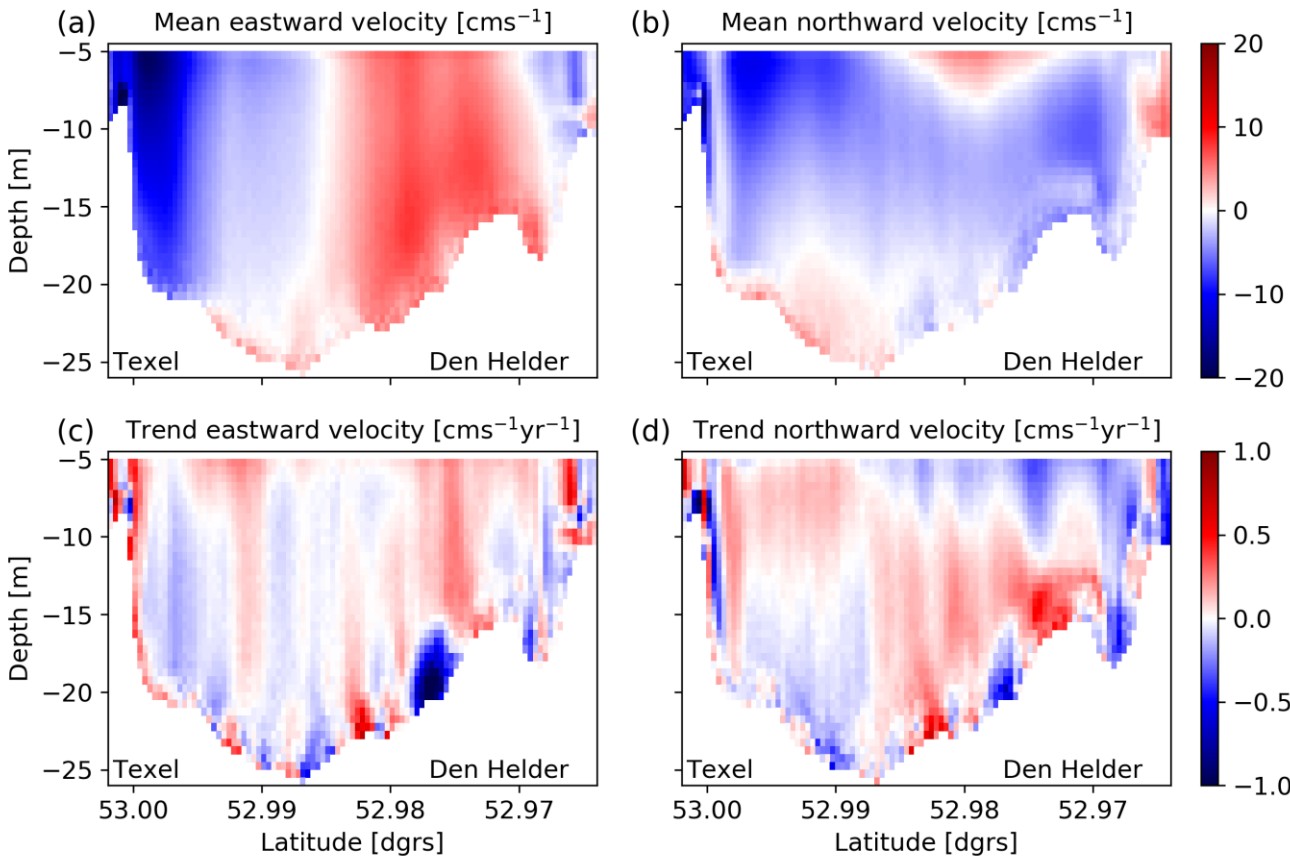

**Figure 7. Long-term mean horizontal velocities in the east a) and north b) direction, and their respective trends c), d). In a) red indicates an in-flow (flood). Levels near the surface subject to vertical tidal changes were excluded.**

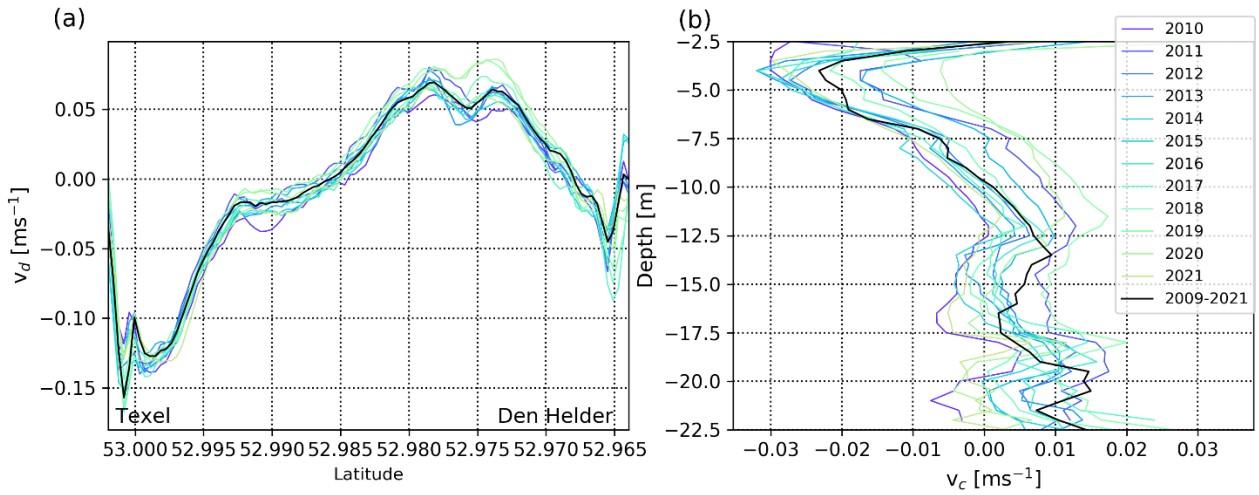





**Figure 8. a) Depth-averaged eastward residual velocities along the transect, and b) cross-section averaged eastward residual velocities as a function of depth for each year, and for the whole time series.**


**Figure 9. Scatter plot of annual mean eastward volume fluxes against annual mean air temperature at 2 m from the ECMWF ERA5 reanalysis.**