# Peer review of "Imminent reversal of the residual flow through the Marsdiep tidal inlet into the Dutch Wadden Sea based on multiyear ferry-borne ADCP observations"

_EGUsphere, 2022_

## Author Response (AR1)

Reviewer 1:
General comments

The paper presents an analysis of the long-term measurements using ADCPs installed on the ferry crossing the Marsdiep tidal inlet in the Dutch Wadden Sea. The analysis finds a significant trend in the residual flow, with decreasing export to the North Sea, and with occasional imports observed in recent years. This is an important finding as the trend has significant effects for the morphodynamics and ecosystem of the western Wadden Sea. Therefore, I support the publication of the paper after proper revision.

A: We thank the reviewer for the time taken to review the manuscript, and for the suggestions and support.

The paper also infers that the trend is caused by the change in the tides in the North Sea related to global warming. For me this part is not yet such convincing that it can be presented as conclusion in the paper. As already mentioned in the paper, at lease the significant change of the O1 component in the horizontal tide should be verified by analyzing the tidal gauge observations. Moreover, it is still not clear to me why larger O1 amplitude causes the observed trend in the residual flow.

A: It was not our intention to suggest that O1 is the main driver for the trend in the residual flow. We merely observe and report that O1 is the only constituent with a consistent and significant trend in the data. We will improve the text to make this clearer.

The correlation between the residual flux and the air temperature does not say much. Given the fact that the temperature had an increasing trend any trend of any process in the same period would probably show a correlation with the temperature. I would suggest two possible revisions: (1) complete the analysis and present more convincing evidence for the conclusion concerning the cause of the observed trend; (2) admit that the cause of the trend is not yet clear and present a serries of possible causes / hypotheses.

A: We agree that our conclusions may be too strong at this stage. We would very much like to and are indeed planning to do a more complete analysis. As this will take considerable effort, and the currently observed trends are new and worth reportin, we consider such analyses one or more separate pieces of work. We will revise the discussion to frame our conclusions more as a hypothesis, and address all potential causes: wind stress (unlikely as already argued), changes in basin morphology, changes in fresh-water discharge (the volume is substantially smaller than historic residual, and can't reverse) and changes in tides. We still think that the latter is the most plausible or dominant cause, but indeed all can contribute. We will remove Figure 9.

Specific comments

Line 43-44: Please specify what type of tidal asymmetry you refer to when saying flood-dominant or ebb-dominant (residual flow, peak velocity, or duration of slack tides?). The same in lines 235-236.

A: We have added this.

Lines 65-66: The previous research did not conclude that the system reached a new dynamic equilibrium, but concluded that the closure of the Zuiderzee still influences the morphological development of the western Dutch Wadden Sea, see also Wang et al. (2018).

A: We think the reviewer is mistaken, or means a different paper than we found. Wang et al. (2018) (Neth. J. Geosci 97, 183-214) make a similar statement as we do in the current manuscript, citing the same source (p. 192, last paragraph in the left column): "the morphodynamic changes in these areas are reaching a nearequilibrium state (see Elias & Van der Spek, 2017, for an analysis of Texel Inlet)".

Lines 181-184: I am curious about the development of the other tidal constituents.

A: Unfortunately, as indicated, O1 was the only constitutent with a statistically significant trend in our analysis. So we can't make meaningful statements about other constituents at this stage. However, as we indicate at the end of Section 4.1, ADCP observations are not the most accurate way to measure changes in tides. We are confident that more detail can be obtained through an in-depth study of the local tide-gauge records (as also stated).

Details

Line 11 (and many more through the paper): "sediment balance"? This term was first presented in Dutch (sedimentbalans) and later it is translated as "sediment budget" in English publications. Personally I think that this is the right translation.

A: Thanks, we will change this.

Line 53: Please change "the waves" to "the sand waves" to avoid confusion.

A: we will change this.

Lines 59-60: Please give more elaboration. The sentence is difficult to follow now.

A: We agree that the sentence is difficult to understand. We will remove the last part 'in agreement with their seasonal variability'.

Lines 138-139: Please check the sentence. It seems suggesting that the flow velocity is used through the whole water depth.

A: we will improve the formulation.

Lines 166: reference error.

A: we will correct this.

Line 177: "residual depth"?

A: we will remove 'residual'.

Lines 253: Please specify who does "They" refer to.

A: we will replace this with the reference: Jänicke e

Reviewer 2:
Review for:

**Imminent reversal of the residual flow through the Marsdiep tidal inlet into the Dutch Wadden Sea based on multiyear ferry-borne ADCP observations**

Based on a long ferry-born ADCP time series of the Marsdiep tidal inlet the authors demonstrate that the residual flow changed from net outflow to net inflow during the last decades. The authors associate this trend to global warming by correlating the yearly net flow to the annual mean air temperature. A couple of potential mechanisms that could yield this behavior are discussed, where a change in tidal amplitude due to increased stratification in the North Sea is suggested.

The data analysis appears to be done very carefully and the manuscript is well-written and easy to read. The main result of the paper, a major shift from out- to inflow in the Marsdiep, is, taken on its own, already worth publishing and quite interesting to the community. When it comes to the discussion of potential mechanisms behind this trend, I still have a couple of questions (see below). Overall, I suggest this manuscript for publication after some minor revisions.

A: We thank the reviewer for the time taken to review the manuscript, and the constructive suggestions.

**General Questions/Comments:**

The argument that a change in tidal amplitude due to increased North Sea stratification causes the change in residual flow is not entirely clear to me. Normally,

the change in tidal range due to stratification is mostly thought to be associated with decreased friction of the semidiurnal tidal constituents (see e.g. Mueller 2012, Graewe 2014). This has to do with the near-resonance of semidiurnal tides with the inertial period in mid-latitude regions, resulting in frictional boundary layers that can extend way beyond the height of the thermocline. Therefore, an increase in vertical stratification can reduce the boundary drag for semidiurnal tidal components. In theory this should be less pronounced for the diurnal tidal constituent. In this context it is surprising that the only trend you observe is in the O1 tidal constituent. In line 255 you mention that this can be explained by the "day/night heating/cooling cycle". This reasoning is not very clear to me. Maybe you could explain a little bit more, what you mean here.

A: Re-reading this, the reviewer is correct. We will remove this sentence, and instead state that it is not clear why the O1 component of the flow changes significantly while the ADCP data could not detect significant changes in other constituents.

The correlation to mean air temperature could also suggest other mechanisms. For instance an increase in the baroclinic pressure gradient between the tidal basin and the North Sea. Due to differential heating the tidal basin could become substantially warmer than the open North Sea, yielding a net baroclinic exchange flow (estuarine circulation). However, it is not obvious how this would yield a change in the net volume flow. The warmer water that leaves the system at the surface gets displaced by colder North Sea water in the deep. Given that this is a two-inlet system could this mean that the deeper of the two inlets (Marsdiep?) needs to accommodate more deep inflow to account for the increase in surface outflow through both inlets? I am not sure if this makes any sense. It is just an idea that came to mind while reading this manuscript.

A: This is an interesting suggestion, but such a mechanism would require climate-induced changes in the difference in air temperature between the tidal basin and the open sea, not just changes in mean air temperature. Moreover, such a change in the spatial temperature gradient would need to be substantial (e.g. a degree or more), which is of the same order of magnitude as the total warming, and hence seems unlikely to have occurred. Moreover, in winter, the tidal basin is colder than the North Sea, further complicating this argument. We will not include this in the manuscript.

I am missing a discussion on the river run off. Isn't there some runoff from the Ijsselmeer? Could it also be that warmer years are also more dry years with less run off, resulting in less net outflow from the basin? Or is the run-off just too small to explain the change? that should be easy to discuss.

A: The runoff was mentioned in the introduction, and is indeed too small. Also in response to a comment by the other reviewer, we will now include and discuss a

more explicit and complete list of potential causes in the discussion, including this one.

In Line 264 you say "that wind-forcing ... is not correlated to the trend observed in the ADCP data". As far as I understand, you only test this for annual mean eastward wind speeds, right? What about strong wind events (storms)? Do you have any way to assess or discuss their potential contribution? You mention in L95 that the ferry does not go during "extreme weather events", could this mean that the observations are biased towards "good weather"? I feel that a brief discussion on the potential effect of storm events could be beneficial.

A: We have indeed only tested this for annual mean eastward wind speeds, as the residual flow is also analysed on such time scales. Indeed, storm events affect the flow on shorter time scales (we will add a remark on this), and such events likely contribute to the variability around the trend. However, there would need to be a systematic change in (annual) storminess if storms were driving the trend, and that would also be expressed in the annual mean winds. Extreme weather events that prevent the ferry from sailing typically occur less than once per year (we will add a remark to this effect in the manuscript) so are very rare and are not likely to affect the observed trend. Wind forcing is now also addressed more explicitly in the discussion.

**Minor:**

Figure 6: The coloring of the lines makes it difficult to distinguish between different years. Could you potentially use a more contrasting coloring? This is also true for figure 8.

A: The color scheme was chosen to be visible to most colourblind people (i.e. avoiding combinations of red, green and brown colours). In most of Figure 6, local trends are systematic, so reading these should be relatively straight forward in the current state. We agree, however, that Figure 8 is less easy to read. We have found a better colour scheme, and will also alternate two line styles between years, which makes it easier to identify the lines for individual years.

Most figures?: Is there a reason why you plot the x-axis (latitude) in reverse. This was a little bit confusing to me. If you don't want to change this. Can you at least mention explicitly in the caption that you are looking into the tidal basin (or in ebb-direction).

A: This is inherent to using a right-hand sided coordinate system, and looking in the positive x-direction (i.e. flood direction, into the basin). We think this results in more intuitive figures than looking in the negative x-direction (ebb direction, out of the

basin) which would result in positive flows coming towards the observer. We will add a sentence about the direction of view to the captions.

Figure 7. I find the double gyre structure in (b) quite interesting. This seems to hint towards an important contribution from lateral circulation to the tidal residual flow. You discuss this a little bit, however (d) suggests that these lateral circulation cells are weakening, which I don't recall to be discussed in the manuscript. I feel a weakening of the secondary circulation could also hint towards potential mechanisms involved in the change of residual flow and are therefore worth to be discussed or at least mentioned.

A: The weakening of the double gyre pattern is mentioned at the end of the second paragraph of Section 3.4. It was indeed overlooked in the discussion. We will add it to the list of changes discussed in Section 4.1, and following Nunes and Simpson (1985) add potential changes in friction to the potential causes.

Figure7c: What is this giant blue blob in panel (c). Is this an artifact or real? In this context I was wondering how much of your transect averaged trend is governed by these strong outliers (also on the edges of the transect)? Is there a way to quantify their contribution, relative to the seemingly more well behaved transect interior?

A: This feature is mentioned in the first paragraph of Section 3.4. We think this is related to the largest observed local change in bathymetry (Figure 6). We will add a remark to this effect. Estimating contributions of these local trends to the over-all trend is less straight-forward than it seems. The calculation of the cross-section resolved results, by necessity, involved a projection, and include near-bed velocity features related to bedforms which migrate on sub-annual time scales, and are also affected by lateral differences in bathymetry between transects. Hence, these near-bed trends in the flow are not very reliable. The calculation of the cross-section aggregated trends avoids these issues by assuming mass conservation, and thus results in a more robust estimate of the overall trend in residual flow than would be achieved by integrating over the cross-section explicit trends depicted in Figure 7. We will add a remark to Section 3.4 stressing that the near-bottom trend estimates have a high level of uncertainty because of these bathymetric differences and changes.

Figure 8b: From this figure it seems as if specifically the residual outflow near the surface has decreased in recent years. Could you comment a little bit on that? Could this be related to less fresh water run-off? or decreased estuarine circulation?

A: Although there are indeed recent years for which this holds, it is difficult to conclude from figure 8 that there is such a trend. We plotted the maximum outflow at -4 m as a function of time and this did not yield a clear trend (we will not include this in the manuscript). As we don't have associated temperature and salinity data,

we lack the information to specifically identify changes in drivers of estuarine circulation.

**Typos**:

L166: something is wrong with the figure reference.

A: We will correct this.

**References:**

Müller, M. (2012), The inï¬‚uence of changing stratiï¬• cation conditions on barotropic tidal transport and its implications for seasonal and secular changes of tides, Cont. Shelf Res., 47, 107–118.

Gräwe et al. (2014), Seasonal variability in M 2 and M 4 tidal constituents and its implications for the coastal residual sediment transport, Geophys. Res. Lett., 41, 5563–5570, doi:10.1002/2014GL060517.